# TBM Hunter: Identify and Score Canonical, Extended, and Unconventional Tankyrase-Binding Motifs in Any Protein

**DOI:** 10.3390/ijms242316964

**Published:** 2023-11-30

**Authors:** Christopher M. Clements, Samantha X. Shellman, Melody H. Shellman, Yiqun G. Shellman

**Affiliations:** 1Department of Dermatology, School of Medicine, University of Colorado Anschutz Medical Campus, Aurora, CO 80045, USA; christopher.m.clements@cuanschutz.edu; 2Department of Computer Science, University of Colorado Boulder, Boulder, CO 80309, USA; samantha.shellman@colorado.edu; 3H. Milton Stewart School of Industrial and Systems Engineering, Georgia Institute of Technology, Atlanta, GA 30332, USA; mshellman3@gatech.edu; 4Charles C. Gates Regenerative Medicine and Stem Cell Biology Institute, School of Medicine, University of Colorado Anschutz Medical Campus, Aurora, CO 80045, USA

**Keywords:** tankyrase (TNKS), tankyrase 2 (TNKS2), poly(ADP-ribosyl)ation (PARylation), PARsylation, poly(ADP-ribose) polymerase (PARP), tankyrase-binding motif (TBM), SASH1, RNF146, anti-cancer drug

## Abstract

Tankyrases, a versatile protein group within the poly(ADP-ribose) polymerase family, are essential for post-translational poly(ADP-ribosyl)ation, influencing various cellular functions and contributing to diseases, particularly cancer. Consequently, tankyrases have become important targets for anti-cancer drug development. Emerging approaches in drug discovery aim to disrupt interactions between tankyrases and their binding partners, which hinge on tankyrase-binding motifs (TBMs) within partner proteins and ankyrin repeat cluster domains within tankyrases. Our study addresses the challenge of identifying and ranking TBMs. We have conducted a comprehensive review of the existing literature, classifying TBMs into three distinct groups, each with its own scoring system. To facilitate this process, we introduce TBM Hunter—an accessible, web-based tool. This user-friendly platform provides a cost-free and efficient means to screen and assess potential TBMs within any given protein. TBM Hunter can handle individual proteins or lists of proteins simultaneously. Notably, our results demonstrate that TBM Hunter not only identifies known TBMs but also uncovers novel ones. In summary, our study offers an all-encompassing perspective on TBMs and presents an easy-to-use, precise, and free tool for identifying and evaluating potential TBMs in any protein, thereby enhancing research and drug development efforts focused on tankyrases.

## 1. Introduction

Tankyrases (TNKS and TNKS2) are members of the poly(ADP-ribose) polymerase (PARP) family, also referred to as PARP5 and PARP6, respectively. Tankyrases modify proteins through poly(ADP-ribosyl)ation (PARylation), which alters protein activity, stability, and subcellular location [1]. Tankyrases play crucial roles in multiple cellular functions, including mitosis, genomic stability, pexophagy, vesicle trafficking, and more [1]. Tankyrases are also well known to regulate cellular signaling pathways including WNT and Hippo, and as such are therapeutic targets for cancers [1,2]. While targeting the enzymatic activity of tankyrases holds promise, it may cause unintended biological consequences due to the large number of diverse substrates, leading to their accumulation as well as that of the tankyrases [3]. Recently, the direct targeting tankyrases scaffolding function has been proposed as a better approach for drug development [3,4,5]. With hundreds of potential diverse binding partners [6], a full understanding of the complexities within their partnerships remains incomplete. The rapid identification and screening of potential binding partners are imperative for gaining deeper insights into the intricate regulatory mechanisms governing the functions of tankyrases.

Tankyrases consist of five ankyrin repeat cluster (ARC) domains, a sterile alpha motif (SAM) domain, and a C-terminal PARP catalytic domain (Figure 1) [1,7]. The SAM domain is involved in polymer formation [8,9,10], the PARP domain is essential for PARylation enzymatic activity, and the ARC domains are responsible for interacting with binding partners [7,11,12]. The binding partners of tankyrases generally have at least one tankyrase-binding motif (TBM), which mediates their interactions via binding to one of the ARC domains in tankyrases [4]. Tankyrases have five ARCs, four of which (ARCs 1, 2, 4, and 5) can bind TBM-containing protein partners [1,4,11,13]. The TBMs in tankyrases’ binding partners are crucial for their functions and their interactions with tankyrases. 

TBMs encompass a diverse array of protein sequences. Canonical TBMs follow a simple 8-residue super consensus, typically characterized by the sequence R-x-x-x-x-G-(no-P)-x, allowing for variations in amino acid preferences at specific positions [11]. In this sequence, “R” represents arginine, “X” can be any amino acid, “P” denotes proline, and “G” represents glycine. This consensus acts as an indicator for potential TBMs, accommodating sequence heterogeneity that enables fine-tuning of the interaction [11].

The tankyrase-targeting scores (TTS) system was developed to assess the binding strength of TBMs to ARCs in tankyrases [11]. A higher TTS score indicates a stronger affinity, while a lower score suggests lower affinity. Nevertheless, TBMs with additional amino acids between R1 and G6 (R1/Gx consensus) have been reported, as seen in mAXIN1 [12] and RNF146 [14]. Moreover, TBMs lacking arginine at position 1 are also possible [4,11,15].

Given the diversity and complexity of these various TBMs, locating and ranking them based on binding strength can be challenging for biologists without bioinformatic skills. Establishing a consensus on classifications for all TBM types and building a searching and evaluating tool for them would greatly assist future studies. This report, based on the literature, categorizes TBMs into three distinct groups, assigns a comparable scoring system for each based on the TTS system, and introduces a user-friendly web-based tool for the rapid screening and assessment of potential TBMs in any protein at https://shellmanlab.github.io/.

## 2. Results

### 2.1. Classification of Three Types of TBMs

The definition of TBMs has undergone continuous refinement over the past several years. Our classification, drawn from the existing literature, identifies three distinct TBM types: canonical, extended, and unconventional TBMs (as outlined in Table 1).

Canonical TBMs adhere to the well-established R1/G6 rule and notably lack proline at position 7, as initially described in [11]. Extended TBMs share similarities with canonical motifs, differing only in the number of residues that separate the R1/Gx residues, as illustrated in [12,14]. Unconventional TBMs represent a unique category where the R1 element is absent, as evidenced by the research [4,15]. Below, we provide a more comprehensive discussion of these distinct TBM categories.

Canonical TBMs represent the most prevalent and extensively studied subset among all TBMs. These include those found in 3BP2 [11], MCL1 [16], NUMA1 [17], and AXIN [12,18]. The consensus sequence for canonical TBMs is a simple 8-residue sequence featuring the R1/G6 motif, as R-x-x-x-x-G-x-x [4,11,12,14]. It is important to note that various canonical TBMs may exhibit different binding affinities to ARCs of tankyrases, which depend on the specific composition of the rest of the sequence in the motif [11]. For instance, research by Guettler and colleagues indicates a preference for the absence of proline at the x7 position, as well as a propensity for acidic side chains at x5 and x7 [11]. Consequently, we have defined canonical TBMs as sequences adhering to the pattern R1-x-x-x-x-G6-(no-P)-x (Table 1).

Extended TBMs, also known as non-canonical TBMs, are less studied [12,14]. Different from canonical TBMs, which exhibit a 4-amino-acid gap between R1 and G6, extended TBMs display an increased number of residues separating the R1/Gx elements. This variation has been verified experimentally in proteins such as RNF146, NELFE, I4FA1, FBOX50, DESM [14], as well as in AXIN1 [12].

Crystallographic studies of the extended TBMs have elucidated that the interaction primarily involves R1, Gx, and the two residues each pre- and post-Gx, while the rest of the amino acids between R1 and Gx function as a flexible loop and do not contribute to the binding [12,14]. More specifically, this interaction is mediated only by R1 along with the two residues immediately preceding and following Gx. These studies have shed light on the potential mechanisms and prerequisites associated with extended TBMs. In recognition of the potential for a high rate of false positive identifications when dealing with greater amino acid extensions between R1 and Gx, we have focused on extended TBMs with one, two, or three additional amino acids, as illustrated in Table 1.

Even rarer are the unconventional TBMs in which the initial position 1 is not occupied by an arginine (R). While arginine at position 1 is required for canonical and extended TBMs, it is worth noting that not all reported TBMs adhere to this criterion. For instance, one such TBM has been validated in EBNA1 (with the sequence **E**GGPD**G**EE) [15] and another is proposed in the case of GRB14 (with the sequence **L**PLPD**G**TR) [4]. 

Crucially, it should be emphasized that G6 remains a vital component of a functional TBM, as any substitutions of G6 in known canonical TBMs [11,12] or in the motif of EBNA1 [15] have been found to abolish their capacity to bind with tankyrases. These atypical TBMs, which lack arginine at position 1, are referred to in this context as “unconventional TBMs” (Table 1).

### 2.2. Building a Web-Based Tool, TBM Hunter, for Identifying and Scoring Various TBMs

In order to facilitate the discovery of TBMs and the assessment of their binding potential within various proteins, particularly for biologists without extensive bioinformatics expertise, we have designed a user-friendly web application known as TBM Hunter, which can be accessed at https://shellmanlab.github.io/.

Our primary objective in creating TBM Hunter was to establish a cost-free and efficient tool that streamlines the process of screening and evaluating potential TBMs in proteins. TBM Hunter accommodates the analysis of individual proteins or multiple proteins simultaneously. This web application features two key functions encapsulated in buttons: “**Find and Score motifs**!” and “**Score any 8-amino-acid sequence**” (Figure 2). TBM Hunter, with its intuitive interface and versatile functionality, empowers researchers to delve into the world of TBMs with ease and precision, serving as a valuable resource for the exploration of protein-binding motifs.

#### 2.2.1. Search Canonical and Extended TBMs with “**Find and Score Motifs**!”

Within TBM Hunter, we have streamlined the process by integrating the search for both canonical and extended TBMs into a single function. This approach is adopted because both categories share a common foundation in the R1/Gx consensus. In contrast, the search for unconventional TBMs poses the risk of producing an excessive number of false positives due to minimal sequence requirements. To address this, we have devised a scoring system for the evaluation of the potential binding strength of any octapeptide sequence, which is elaborated upon in Section 2.2.2. This design enhances the precision and effectiveness of TBM Hunter, ensuring that researchers can efficiently assess a wide range of binding motifs.

To identify TBMs conforming to the criteria established for canonical TBMs, as outlined in Table 1, the “**Find and Score Motifs**!” function employs a three-step process. It begins by searching for the presence of an arginine (R) and subsequently checks for a glycine (G) at the R + 5 position, while also confirming the absence of proline (no-P) at R + 6. These conditions are applied consistently, regardless of the specific residue content in the remaining portion of the 8-amino-acid sequence.

In the case of extended TBMs, the “**Find and Score Motifs**!” function follows a similar methodology as for canonical TBMs, with the sole exception being the allowance of one, two, or three additional residues between the arginine and glycine positions. In this instance, the tool searches for the presence of a glycine at R + 6, R + 7, and R + 8 to identify potential extended motifs, effectively accommodating variations in their sequence composition.

#### 2.2.2. Score TBMs with “**Find and Score Motifs**!” and “**Score Any 8-Amino-Acid Sequence**”

While arginine (R1) and glycine at the Gx position serve as crucial amino acids enabling the binding of canonical and extended TBMs to tankyrases, it is noteworthy that the composition of the remaining sequence of TBMs plays a significant role in influencing their binding affinities [11]. Therefore, having a scoring system to assess the binding strength of the potential TBMs identified through TBM Hunter is invaluable for researchers, allowing them to prioritize candidate TBMs for subsequent validation experiments.

In our evaluation of the relative binding strength of the identified TBMs, we have adopted the tankyrase-targeting score (TTS) system, originally devised by Guettler and colleagues for quantifying the binding affinity of potential TBMs [11]. To gauge how effectively a peptide binds to tankyrases, they constructed a position-specific scoring matrix (PSSM) to compute the TTS of a peptide sequence. This system was developed through an exhaustive analysis of a peptide library that originated from 3BP2’s TBM, wherein each of the eight amino acid positions were systematically replaced with all twenty amino acids. The PSSM assigns a score to each amino acid at every position in the peptide sequence. The TTS is subsequently calculated by summing these scores, with a higher TTS value indicating a stronger binding affinity. For added context, the TTS is normalized by the maximum achievable value to account for varying peptide sizes.

Given their shared octapeptide length, we employed the same scoring system for canonical and unconventional TBMs. For extended TBMs, to address the potential loop region introduced by the extra amino acid, we have adapted a strategy akin to that described in [11] for peptides of different sizes (further details in the Materials and Methods section).

This approach facilitates the identification of sequences that may function as TBMs, enables in silico predictions of the impact of residue substitutions, and quantifies the design of TBM mimetics or peptide-based inhibitors, offering valuable insights for further research and drug development.

### 2.3. TBM Hunter Identified and Scored Known Canonical TBMs

To assess the performance of our tool in detecting canonical TBMs, we selected proteins known to contain experimentally validated TBMs, and the results are shown in Table 2 and Appendix A. These proteins encompass several established tankyrase-binding partners, supported by prior studies [4,11,19], along with a negative control, AMPK, which has been confirmed to not possess a TBM or exhibit binding to TNKS [19]. To maintain robust control over the test, we opted for LKB1 and AMPK as our positive and negative controls, respectively, given that both were examined in the same research study [19]. Experimentally validated TBMs are indicated in bold and underlined for clarity.

It is worth noting that our tool, due to its high sensitivity, often identified multiple TBMs within each protein. Notably, AXIN1 exhibited the lowest TTS score among the confirmed TBMs in Table 2, registering at 0.38, while some unverified peptides displayed significantly higher scores, such as TERF1 at 396 with a TTS of 0.76. The results presented herein provide strong validation for the effectiveness of our tool in detecting canonical TBMs.

### 2.4. TBM Hunter Identified and Scored Known Extended TBMs

To evaluate the functionality of our extended TBM detection tool, we conducted searches within proteins known to contain validated extended TBMs, specifically RNF146, NELFE, and I4FA1 (as detailed in Table 3). While each protein revealed multiple extended TBMs during the identification phase, only those that had been previously confirmed to bind to tankyrases exhibited TTS scores exceeding 0.5 [14]. Some peptide motifs did score below this threshold but have been validated, as exemplified by the RNF146 TBMs at positions 220 and 333. Additionally, it is worth noting that applying the 0.38 cutoff observed for canonical motifs in Table 2 raises the possibility of IF4A1 harboring a second extended TBM at position 45 (Table 3).

Although TTS scores for extended TBMs are expected to exhibit a level of comparability with canonical TBMs, this has yet to be confirmed experimentally through a peptide library analysis. Consequently, any comparisons between these two TBM types should be approached with a degree of caution.

### 2.5. TBM Hunter Scored Unconventional TBMs

To assess the effectiveness of our scoring system in scoring unconventional TBMs, we focused on two previously reported cases: EBNA1 (EGGPDGEE) [15] and GRB14 (LPLPDGTR) [4] (Table 4). Notably, these unconventional TBMs obtained scores of 0.44 and 0.38, respectively, both exceeding or equal to 0.38. This places them comfortably within the range of validated TBMs as presented in Table 2. Therefore, these scores provide compelling evidence of their legitimacy as genuine TBMs.

### 2.6. TBM Hunter Discovered Novel TBMs in SASH1, Which Were Experimentally Validated

Excitingly, our tools have yielded new and bona fide TBMs, a significant achievement documented in a recent study [20]. In this research, we employed yeast two-hybrid screening to identify TNKS2 as a potential binding partner for SASH1. By harnessing TBM Hunter, we successfully discovered multiple prospective TBMs, with a particular emphasis on the SPIDER region of SASH1. Among these candidates, three exhibited a high degree of conservation across vertebrates. Subsequently, using two-dimensional ^1^H-^15^N HSQC experiments involving the titration of TNKS2-ARC4 into ^15^N-labeled SPIDER, it was empirically verified that all four predicted canonical TBMs indeed interacted with the ARC4 domain of TNKS2 (Table 5). This success story illustrates the efficacy of our tools in identifying novel TBMs, offering a user-friendly solution that eliminates the requirement for advanced computational skills.

## 3. Discussion

TBMs play a pivotal role in mediating interactions between tankyrases and their binding partners. However, the discovery of new TBMs and the precise identification of their locations can be challenging, often requiring extensive computational expertise. Moreover, a distinct lack of readily accessible tools for ranking identified TBMs has posed a hurdle in this area of research. To address this, we offer comprehensive guidelines for the classification and identification of TBMs. In addition, we introduce TBM Hunter, a user-friendly web-based toolkit designed to simplify the process of TBM identification and ranking. TBM Hunter goes a step further by providing a preliminary quantification of TBM binding strength through the implementation of a scoring system based on the rigorous TTS scoring criteria, which extends its applicability to encompass extended motifs. Furthermore, we have incorporated a feature that enables the scoring of any octapeptide according to the TTS criteria, facilitating comparisons between suspected unconventional TBMs and known or predicted TBMs.

TBM Hunter holds immense potential for diverse applications in biological research. It proves particularly valuable in cancer research, drug development, and structural and cellular biology studies focused on tankyrases and their interacting partners. Given the wealth of omics data available, our tool serves as an invaluable resource for researchers exploring various gene/protein datasets related to the study of tankyrases and their binding partners. For example, it expedites the identification of potential TBMs, shedding light on the intricate network of tankyrase interactions, thereby enhancing the understanding of these interactions for further research.

Notably, tankyrases are implicated in multiple cancer types and other diseases, making them attractive therapeutic targets. Our tool bridges the gap between tankyrases and their binding partners, offering a pathway to manipulate these interactions for therapeutic benefit. In the realm of drug development, TBM Hunter facilitates in silico predictions of the impact of residue substitutions, providing quantitative comparisons that inform the design of TBM mimetics or peptide-based inhibitors, thus offering valuable insights for pharmacological research.

It is essential to bear in mind that our tools are designed for high sensitivity, which does introduce the potential for false positives. Researchers should carefully consider both the location and score of the predicted TBMs, before proceeding to experimentation. TBMs should be located in structurally disordered regions to adopt the required conformation for binding, in line with the established research [11,14]. Protein structures from the Protein Data Bank (https://www.rcsb.org/) or structural predictions from resources like Google’s AlphaFold (https://alphafold.ebi.ac.uk/; also accessible in UniProt) [21] can assist in assessing the relevance of sequence-based findings. Additionally, it is worth noting that Guettler et al. utilized stringent cutoffs including ≥0.77 in their study [11], a value that may be overly stringent in some cases, considering their exclusion of the bona fide TBM of SASH1 aa512-519 that our tools identified [20]. Furthermore, Pollock et al. have outlined an excellent experimental workflow for validating tankyrase binders [4], offering valuable guidance for researchers in this field.

We need to acknowledge that the TTS system was developed through individual substitutions of each of the 20 common amino acids into every position of the 3BP2 TBM peptide [11]. When applied to other motifs, this methodology does not fully account for the potential influence of the nearest-neighbor effect, wherein interactions between residues (*i* to *i* ± 1 interactions) within the peptide may impact binding. Changes in one residue side chain (*i*) can directly affect those immediately adjacent to it (*i* ± 1) through chemical interactions or rotational constraints [22]. This effect could propagate across the peptide, introducing or removing additional chemical and spatial constraints, unaccounted for in the TTS system. This may elucidate why certain specific residues, disallowed at particular positions in the TTS system, may be present in other favorable binders, such as the isoleucine at position 5 of the MCL1 TBM (starting residue R78), despite its unfavorable ranking in the TTS system at that position [11]. Additionally, the TTS system was built upon data of TBM peptides interacting with the ARC4 domain of TNKS2 [11], implying potential variability in experimental binding outcomes depending on the utilized ARC domain. Moreover, the theoretical maximum score is 1 [11]. The inclusion of an R1 or G6 adds a score of +0.26 each to the overall score, while any other amino acid at these positions scores −0.13. This amounts to a total change of 0.39 for either R1 or G6 in comparison to any other amino acid at these positions [11].

## 4. Materials and Methods

### 4.1. Computer Program Languages and the Devleopment of TBM Hunter

The TBM Hunter website was developed using Github Pages, and all the code is freely accessible at https://github.com/ShellmanLab. The primary coding language used is JavaScript, with supplementary code in HTML and CSS. Visual Studio IDE (Version: Community 2022) was employed for code editing.

### 4.2. Protocol for Identifying and Scoring Canonical and Extended TBMs Using the “Find and Score Motifs!” Function

TBM Hunter, the web application is freely available at https://shellmanlab.github.io/. A comprehensive protocol is available at Protocols.io [23]. In brief, users entered the UniProt codes for desired proteins into the first text box on TBM Hunter and clicked the “Find and Score Motifs!” button. The results were displayed at the bottom of the webpage, copied, and pasted into an Excel spreadsheet, then comma-delimited into individual cells for organization and analysis.

### 4.3. Protocol for Scoring Unconventional TBMs Using the “Score Any 8-Amino Acid Sequence” Function

A detailed protocol can be found at Protocols.io [23]. In brief, users entered the octapeptide of interest in all caps into the second textbox and selected “Score any 8-amino acid sequence.” The resulting score was displayed at the bottom of the webpage.

### 4.4. Scoring Systems for TBMs

Our scoring systems are based on the TTS developed for canonical TBMs as in [11]. For scoring either a canonical TBM or any octapeptide, minor adjustments were made to simplify coding without affecting the final TTS value. The scores presented in Appendix A are derived from the scores in Table S7 of [11], divided by 3.86, the maximum possible sum of the scores for an octapeptide. The final normalized score has a theoretical range from a minimum of −0.77 to a maximum of 1, where higher scores indicate a greater likelihood of binding. The normalized scoring system is calculated according to Equation (1):(1)$$TSS=\frac{{PSSM}_{pos.1}}{max\left(\sum_{pos.=1}^{n}{PSSM}_{pos.}\right)}+\ldots +\frac{{PSSM}_{pos.n}}{max\left(\sum_{pos.=1}^{n}{PSSM}_{pos.}\right)}=\sum_{pos.=1}^{n}{PSSMnorm}_{pos.}$$
where *n* = 8, *PSSM_pos_* (position-specific scoring matrix) represents the value for a specific position in the matrix, and *PSSMnorm_pos_* is the normalized PSSM at each position.

The scoring system for extended TBMs is similar to that of canonical TBMs. However, because only limited residues are involved in binding, only those positions will be scored as seen in Table 1. Thus, we removed positions 2 and 3 from the octapeptide matrix for canonical TBMs, and this yielded a theoretical maximum value of 3.63 for normalization, and the resulting score matrix for extended TBMs is shown in Appendix A. This normalization yields a score ranging from −0.69 to 1.00, in contrast with the original TTS system for hexapeptides that excluded positions 7 and 8 [11]. 

## 5. Conclusions

Our web-based tool, TBM Hunter, offers a user-friendly and invaluable resource for researchers delving into the study of tankyrases and their interacting partners. TBM Hunter is freely accessible and easy to use, and it can be found at https://shellmanlab.github.io/. The procedures for utilizing it are straightforward—researchers need only input the UniProt code(s) of protein interest(s) and click the relevant tool button. The output includes the identification of sequences, positions, and scores of the potential TBMs in the searched proteins. Our tools have not only successfully validated known TBMs but have also revealed new, bona fide TBMs. As such, these tools offer immense value to both those embarking on the exploration of classical tankyrase partners and researchers seeking to further characterize potential novel interactions with tankyrases, as well as drug development. We believe that this will serve as a model for the development of future tools aimed at detecting other binding motifs important in health and disease.

## Figures and Tables

**Figure 1 ijms-24-16964-f001:**
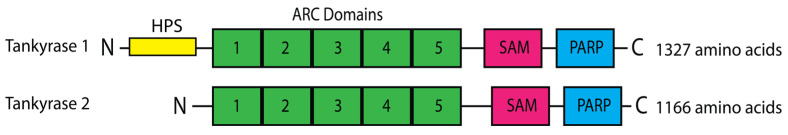
Tankyrases are multifunctional proteins with scaffolding capacity on the N-terminal and an enzymatic PARP domain at the C-terminal.

**Figure 2 ijms-24-16964-f002:**
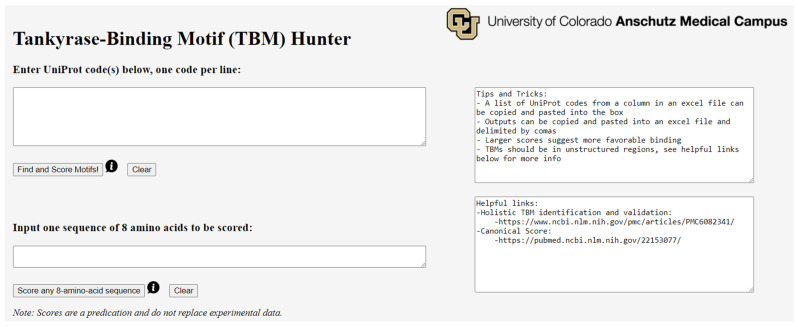
Homepage of the TBM Search Project (TBM Hunter) at https://shellmanlab.github.io/. The webpage is divided into two sections: the left side is the input and output areas for the two function buttons, “**Find and Score motifs**!” and “**Score any 8-amino-acid sequence**”. The right side provides the relevant information and references. Both buttons operate independently, with their results displayed at the bottom of the webpage.

**Table 1 ijms-24-16964-t001:** TBM identification and scoring rules. Amino acid positions used for scoring are illustrated in red.

	Sequence	Scoring
Canonical TBM	R1-x-x-x-x-G6-(no-P)-x	All residues
	R1-x-x-x-x-x-G7-(no-P)-x	
Extended TBM	R1-x-x-x-x-x-x-G8-(no-P)-x	R1, *i* − 2, *i* − 1, *i*, *i* + 1, *i* + 2 (*i* = G7, G8, or G9)
	R1-x-x-x-x-x-x-x-G9-(no-P)-x	
Unconventional TBM	x-x-x-x-x-G6-x-x	All residues

**Table 2 ijms-24-16964-t002:** Canonical TBMs. For easy viewing, only peptides with TTS values above 0.35 are shown here. The entire table can be found in Appendix A.

Protein	UniProt Code	Motif Seq	Starting Position	Score	References
3BP2	P78314	** RSPPDGQS **	** 415 **	** 0.80 **	[11]
MCL1	Q07820	RNAVIGLN	6	0.45	
		RREIGGGE	45	0.35	
		REIGGGEA	46	0.65	
		** RPPPIGAE **	** 78 **	** 0.69 **	[11,16]
		REQATGAK	187	0.48	
		RRVGDGVQ	214	0.89	
NUMA1	Q14980	RAEELGQE	1321	0.37	
		** RTQPDGTS **	** 1743 **	** 0.79 **	[11,17]
		RDRHEGRK	2017	0.46	
AXIN1	O15169	** RPPVPGEE **	** 22 **	** 0.38 **	[11,18]
		RRYSEGRE	284	0.49	
		RMEEEGED	417	0.54	
		RRTGHGSS	647	0.61	
		RKVGGGSA	770	0.61	
TERF1	P54274	** RGCADGRD **	** 13 **	** 0.78 **	[11,17]
		RAFRDGRS	88	0.45	
		RKYGEGNW	396	0.76	
FAT4	Q6V0I7	RLQDEGTP	288	0.48	
		RCVPPGDC	4423	0.49	
		** RKQPEGNP **	** 4572 **	** 0.64 **	[11]
		RHSPLGFA	4653	0.49	
		** RNPADGIP **	** 4827 **	** 0.76 **	[11]
DISC1	Q9NRI5	RARQCGLD	82	0.48	
		RVRAAGSL	170	0.46	
		** RGEAEGCP **	** 223 **	** 0.65 **	[11]
		REGLEGLL	618	0.47	
BABA1	Q9NWV8	** RSNPEGAE **	** 28 **	** 0.70 **	[11]
		** RSEGEGEA **	** 48 **	** 0.82 **	[11]
LKB1	Q15831	** RAKLIGKY **	** 42 **	** 0.44 **	[19]
		** RRIPNGEA **	** 86 **	** 0.50 **	[19]
AMPK	P54619	N/A	N/A	N/A	[19]

**Table 3 ijms-24-16964-t003:** Extended TBMs. Experimentally validated TBMs are marked with bold and underlining.

Protein	UniProt Code	Extended TBM Seq	Starting Position	Score
RNF146	Q9NTX7	** RESSADGAD **	** 194 **	** 0.54 **
		** RPLTSVDGQL **	** 220 **	** 0.34 **
		** RSHRGEGEE **	** 260 **	** 0.59 **
		** RSVAGGGTV **	** 333 **	** 0.35 **
		** RSRRPDGQC **	** 346 **	** 0.53 **
IF4A1	P60842	** RSRDNGPDGME **	** 8 **	** 0.57 **
		RDNGPDGME	10	0.57
		RGIYAYGFE	45	0.38
		RAILPCIKGYD	61	0.06
		RENYIHRIGRG	353	0.17
		RIGRGGRFGRK	359	0.01
		RGGRFGRKGVA	362	0.03
NELFE	P18615	RSRTLEGKL	88	0.19
		RLRELGPDGEE	142	0.59
		** RELGPDGEE **	** 144 **	** 0.59 **
		RDRDRDREGPF	236	0.04
		RDRDREGPF	238	0.04
		RRAPRKGNT	255	0.02
		RGAFSPFGNI	277	0.21

**Table 4 ijms-24-16964-t004:** Unconventional TBMs.

Protein	Motif Seq	Score
EBNA1	EGGPDGEE	0.44
GRB14	LPLPDGTR	0.38

**Table 5 ijms-24-16964-t005:** Predicted and validated TBMs in SASH1.

Starting Position	Motif Seq	Score	Type	Validation
404	RTCSFGGF	0.30	Canonical	Yes
416	RSLHVGSN	0.46	Canonical	Yes
512	RSSLSGQS	0.33	Canonical	Yes
552	RGPFCGRA	0.46	Canonical	Yes

## Data Availability

All data are shown in this manuscript, with all the computer code freely accessible at https://github.com/ShellmanLab.

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
