# Peer review of "TBM Hunter: Identify and Score Canonical, Extended, and Unconventional Tankyrase-Binding Motifs in Any Protein"

_ijms, 2023, doi:10.3390/ijms242316964_

Round 1
Reviewer 1 Report
Comments and Suggestions for Authors
Dear Authors,
please add abbreviations list to the text flow. In the keywords list introduce full names of the abbreviated words. Correct text spelling in line 378. In the References, Ref.3 should contain page and volume numbers. Referenced journal article names explain in small letters, except the first word.
- The main question is to identify and score canonical, extended, and unconventional tankyrase-binding motifs (TBMs)in any protein, using DB TBM Hunter.
- MS text is relevant in the field. It extends and broudens the TBM Hunter use area.
- Results shown in MS demonstrate that TBM Hunter not only identifies known TBMs but also uncovers novel ones.
- The methodology is appropriate.
- The conclusion is,that this study offers an all-encompassing perspective on TBMs and presents an easy-to-use, precise, and free tool for identifying and evaluating potential TBMs in any protein, thereby enhancing research and drug development efforts focused on tankyrases. Thus the conclusions are consistent with the evidence and arguments presented and they do address the main question posed.
-The references are appropriate.
-Tables and figures are appropriate.
Reviewer 2 Report
Comments and Suggestions for Authors
1) First of all, I would like to congrats the authors for such nice study, mostly regarding its valuable TBM hunter. I have tested the TBM hunter for some specific proteins and I just delighted me with its user-friendly interface (input and output areas), mainly for it be a web-based and cost-free tool. I have an only question for the authors: Do you think would be possible "adapting" it or develop a similar tool to search and scoring other motifs for different protein families?
